# Statistical and Network-Based Analysis of Italian COVID-19 Data: Communities Detection and Temporal Evolution

**DOI:** 10.3390/ijerph17124182

**Published:** 2020-06-12

**Authors:** Marianna Milano, Mario Cannataro

**Affiliations:** Data Analytics Research Center, Department of Medical and Surgical Sciences, University of Catanzaro, 88100 Catanzaro, Italy; m.milano@unicz.it

**Keywords:** COVID-19, network analysis, community detection

## Abstract

The coronavirus disease (COVID-19) outbreak started in Wuhan, China, and it has rapidly spread across the world. Italy is one of the European countries most affected by COVID-19, and it has registered high COVID-19 death rates and the death toll. In this article, we analyzed different Italian COVID-19 data at the regional level for the period 24 February to 29 March 2020. The analysis pipeline includes the following steps. After individuating groups of similar or dissimilar regions with respect to the ten types of available COVID-19 data using statistical test, we built several similarity matrices. Then, we mapped those similarity matrices into networks where nodes represent Italian regions and edges represent similarity relationships (edge length is inversely proportional to similarity). Then, network-based analysis was performed mainly discovering communities of regions that show similar behavior. In particular, network-based analysis was performed by running several community detection algorithms on those networks and by underlying communities of regions that show similar behavior. The network-based analysis of Italian COVID-19 data is able to elegantly show how regions form communities, i.e., how they join and leave them, along time and how community consistency changes along time and with respect to the different available data.

## 1. Introduction

Coronavirus disease, known as COVID-19, emerged in the city of Wuhan, in China, in November 2019 [1].

The disease is caused by the novel coronavirus Sars-CoV-2 [2] and its clinical manifestations include fever, cough, fatigue, chest distress, diarrhea, nausea, vomiting [3] and also acute respiratory distress syndrome in severe cases [4].

COVID-19 is characterized by a long incubation period, high infectivity, and different transmission methods [5]. The contagion happens mainly through respiratory and blood contact with the coronavirus.

In a few months, COVID-19 epidemic quickly spread to Asian countries and it reached more than 200 countries in the world, causing tens of thousands of deaths.

On 11 March 2020, COVID-19 disease was declared a pandemic by the World Health Organization.

In Italy, COVID-19 was identified in January 2020 [6] and the outbreak started in Lombardi and Veneto at the end of February 2020. From the northern regions of Italy, the disease spread very quickly to the nearest regions and then to the rest ones. Italy was considered one of the main epicenters of the pandemic, with 97,689 infections and 10,799 deaths up to 29th of March. The aim of this study is to provide a graph-based representation of daily data provided by Italian Civil Protection that enables evaluation of which regions show similar behavior and discovery of communities. The data refers to the period 24 February to 29 March 2020. To do this, we designed an analysis pipeline to model Italian COVID-19 data as networks and to perform network-based analysis. At first, for each type of data, we evaluated the similarity among a pair of regions by using statistical tests, and accordingly, we built ten similarity matrices (one for each Italian COVID-19 datum). After that, we mapped the similarity matrices into networks where the nodes represent the Italian region, and the edges connect statistically similar regions. Finally, we evaluated how the networks evolved over the weeks by analyzing the networks at different time points: (i) over the period 24 February to 29 March 2020 (study period); and (ii) in single weeks. Then, network-based analysis was performed mainly to discover communities of regions that show similar behavior. The main contribution of the paper is a network-based representation of COVID-19 diffusion similarity among regions and graph-based visualization to underline similar diffusion regions.

The rest of the paper is organized as follows: Section 1 presents the pipeline to analyze Italian COVID-19 data, Section 2 presents the application of our methodology on Italian COVID-19 data, and Section 3 discusses the results. Finally, Section 4 concludes the paper.

## 2. Analysis Pipeline

We designed an analysis pipeline with the goal of investigating similarity among Italian regions with respect to data provided by Italian Civil Protection and to identify clusters of regions with similar behavior.

The analysis pipeline includes the following steps:Building of a similarity matrix. The first step consists of the building of a similarity matrix that records the similarity among a pair of regions with respect to an Italian COVID-19 data measure. The similarity is computed by applying a statistical test. We decided to use the Wilcoxon Sum Rank Test. Therefore, the (i, j) value of the matrix for data k (e.g., swab data) represents the *p*-value of the Wilcoxon statistical test obtained by performing the test on the swab measures of region i with respect to region j. Lower *p*-value means that regions are more dissimilar with respect to that measure. Higher *p*-value means that regions are more similar with respect to that measure. We used the usual significance threshold of 0.05, thus matrices report only *p*-vales ≥ 0.05, while *p*-values < 0.05 are mapped to zero.Mapping similarity matrices to networks. The second step consists of the building networks starting from the similarity matrices. We map each matrix M(i, j) to a network N, where nodes represent the Italian regions and an edge connects two regions (i, j) if the *p*-value in the similarity matrix is greater than the significance threshold of 0.05. edges are weighted with the *p*-value.Temporal analysis of networks. The third step consists of the building of the network at different time intervals. Assuming that the analyzed data presents a temporal evolution, for each one, the corresponding networks at different time points (i.e., at the end of week 1,2, …, 5) and for an study period are built.Community detection. The fourth step consists of the extraction of community on the network by applying an appropriate community detection algorithm. For each network, we extracted subgroups of regions that form a community based on similarity of point of view. The identification of community is performed on the networks related to the study period and for all single week. Then, we extract the communities at different time points, i.e., at the end of the first week, after three weeks, and after five weeks (the study period).

## 3. Results

We applied the designed pipeline to analyze the data at different temporal zoom levels e.g., by analyzing the period from 24 February to 29 March 2020 and by focusing on single weeks as well as the entire observation. For convenience, in the rest of paper we refer to the period 24 February to 29 March 2020 as the study period.

### 3.1. Input DataSet

The present analysis was carried out on the dataset of COVID-19 updated at the https://github.com/pcm-dpc/COVID-19 database, provided by Italian Civil Protection. The dataset consists of:Hospitalized with Symptoms, the numbers of hospitalized patients that present COVID-19 symptoms;Intensive Care, the numbers of hospitalized patients in Intensive Care Units;Total Hospitalized, the total numbers of hospitalized patients;Home Isolation, the numbers of subjects that are in isolation at home;Total Currently Positive, the numbers of subjects that are coronavirus positive;New Currently Positive, the numbers of subjects that are daily coronavirus positive;Discharged/Healed the numbers of subjects that are healed from the disease;Deceased, the numbers of dead patients;Total Cases, the numbers of subjects affected by COVID-19;Swabs, the numbers of test swab carried on positive subjects and on subjects with suspected positivity.

The data is daily provided for each Italian region. The data occupies 47.6 Mbytes of memory.

### 3.2. Building of Similarity Matrices

To build similarity matrices for Italian COVID-19 data, we performed a set of statistical analyses. All analyses are performed by using R software [7]. At first, we computed the main descriptive statistics for all regions in the study period, reported in Table 1.

Figure 1 conveys the evolution of all datasets over days.

After that, we analyze the data evolution by focusing on each single week:The first week starts on 24 February and ends on 1 March;The second starts on 1 March and ends on 8 March;The third starts on 9 March and ends on 15 March;The fourth starts on 16 March and ends on 22 March;The fifth starts on 23 March and ends on 29 March.

As a preliminary test, we applied Pearson’s chi-square test. The *p*-value was less than 0.05 for each distribution data, i.e., data was not normally distributed. According to this, we performed the paired comparison and multiple comparison of data by using two non-parametric tests: Wilcoxon Sum Rank test and Kruskal–Wallis test.

#### 3.2.1. Wilcoxon Sum Rank Test

As an initial step, we used the Wilcoxon Sum Rank test to carry out an analysis within the same type of data for all weeks and then, for each single week. The Wilcoxon test is a non-parametric test designed to evaluate the difference between two treatments or conditions where the samples are correlated. We applied the Wilcoxon test to perform a pair-wise comparison among regions with the goal of highlighting statistically similar distributions among them. For this reason, we built a similarity matrix for each couple of regions, for each of the available COVID-19 data. Figure 2 reports the heat map of similarity value related to Hospitalized with Symptoms network for all regions in the study period. We reported the heat maps for Italian COVID-19 data in the study period in Appendix A, for the lack of space. In addition, we report the Tables of the similarity values computed for Italian COVID-19 data in the study period and in the single weeks in Appendix A.

Results show that according to the type of data, a significant difference exists (*p*-value less than 0.05) among some regions while for others, it is possible to highlight statistically similar distributions. Also, the significance varies by performing the analysis on whole selected time interval and on single week.

#### 3.2.2. Kruskal–Wallis Sum Rank Test

After that, we used the Kruskal–Wallis test performing an analysis on the same type of data for all regions (i.e., carrying out multiple comparisons) for the study period and then, for each single week. The Kruskal–Wallis test is a non-parametric method for analysis of variance used to determine if more samples originate from the same distribution. The results confirmed a significant difference considering all regions on the same type of data for the study period for every single week.

#### 3.2.3. Multiple Linear Regression

Furthermore, we performed multiple linear regression by considering nine indicators: Hospitalized with Symptoms, Intensive Care, Home Isolation, Total Currently Positive, Discharged/Healed, Deceased, Total Cases, Swabs and two geographic factors: population density and number of intensive care beds for regions. We perform a standardization of variables as a preprocessing step. Data related to population density and intensive care beds for regions are reported in Table 2. According to multiple linear regression, we built nine models for each piece of Italian COVID-19 data in order to evaluate an outcome of each indicator on the basis of multiple distinct predictor variables i.e., Population Density and Intensive Care Beds. Table 3 reports the *p*-values associated with the Population Density and Intensive Care Beds and the Multiple R-squared. It is possible to notice that the intensive care beds variable is significantly related to Hospitalized with Symptoms, Intensive Care Home Isolation, Total Currently Positive, Discharged/Healed, Deceased, Total Cases variables with Multiple R-squared greater than 0.5. Instead, population density is significantly related to the Swabs variable. In this case, Multiple R-squared is equal to 0.318 (i.e., 32% of the data is explained by the explanatory variable). These results demonstrate that the population density does not influence Hospitalized with Symptoms, Intensive Care Home Isolation, Total Currently Positive, Discharged/Healed, Deceased, Total Cases variables which can be affected by other factors such as smog or climate as reported in [8,9].

### 3.3. Mapping Similarity Matrices to Networks

To evaluate the evolution of Italian COVID-19 data and evidence which regions show similar behavior, we built networks of each piece of data [10] starting from the result of Wilcoxon test. The nodes of the networks are the Italian regions and the edges link two regions (nodes) with similar trend according to significance level (*p*-value > 0.05) obtained from the Wilcoxon test, otherwise (*p*-value < 0.05) there is not connection among nodes. The network analysis is performed using the igraph libraries [11].

At first, we built ten networks, one for each data (Hospitalized with Symptoms, Intensive Care data, Total Hospitalized, Home Isolation, Total Currently Positive, New Currently Positive, Discharged/Healed, Deceased, Total Cases, Swab) by considering the period 24 February to 29 March. Then, we built the same networks by considering single weeks. The ten networks for the study period and for five weeks are reported in Figure 3, Figure 4, Figure 5, Figure 6, Figure 7, Figure 8, Figure 9, Figure 10, Figure 11 and Figure 12.

### 3.4. Community Detection

Starting from the ten networks related to all five weeks, we wanted to identify which regions form a community from the similarity point of view. For this, we applied the Walktrap community-finding algorithm [12] that identifies densely connected subgraphs, i.e., communities, in a graph via random walks. The idea is that short random walks tend to stay in the same community.

The extracted communities from all Italian COVID-19 networks in the study period are reported in Figure 13.

## 4. Discussion

Figure 3 shows the evolution of Hospitalized with Symptoms network over five weeks. Figure a reports the network that represents the behavior of regions respect to number of hospitalized patients up to 29 March, whereas Figure 3b–f represent the networks on each single week. It is possible to notice that the network structure changes according to the analyzed time interval. At the end of the 35th day, the network has all nodes connected with exception of a single node that represents the Basilicata region. Furthermore, it is possible to highlight different community structures consisting of groups of regions with similar trends. Figure 13a reports five communities: the first one consists of Basilicata; the second one is composed by Piemonte, Marche, Emilia, Lombardi, Veneto; the third one is composed by Liguria, Lazio, Toscana; the fourth one is formed by Campania, Puglia, Sicilia, Abruzzo, Valle d’Aosta, Friuli, Trento, Bolzano; the last one is composed by Umbria, Sardegna, Calabria, Molise.

Figure 4 represents the evolution of the Intensive Care network. It is possible to notice that Lombardi and Veneto, which are the regions most affected by coronavirus disease with the highest numbers hospitalized in Intensive Care Units, are disconnected in the first week. In the second week, Lombardi and Veneto are connected by an edge that represents a level of similarity, whereas in the fifth week Veneto is linked with Emilia and Lombardi and this group of regions becomes a disconnected component among other regions. Thus, while initially Lombardi and Veneto showed a similar trend of Intensive Care data, after Veneto moved far from Lombardi trend. Furthermore, by analyzing the communities in the Intensive Care network (Figure 13b), it is possible to demonstrate four subgraphs formed by (i) Lombardi and Veneto, (ii) Umbria and Lazio, (iii) Marche, Emilia, Piemonte, Toscana and (iv) a large module formed by Campania, Sicilia, Sardegna, Abruzzo, Umbria, Calabria, Basilicata, Bolzano, Valle d’Aosta, Friuli, Trento, Molise.

Figure 5 shows the Total Hospitalized network evolution. Starting form the first week, the structure of the network has two disconnected nodes representing Lombardi and Veneto, whereas, in the second week, the network evolves by presenting a single disconnected node that represent Emilia, and two connected nodes, Lombardi and Veneto, which in turn are disconnected from dense subgraph. In the third and fourth weeks, the network structure presents all connected components, and a high number of nodes are disconnected in the fifth week. Finally, all regions are connected in the final network. By analyzing the communities detected in Total Hospitalized network (Figure 13c), it is possible to notice a similarity with respect to those extracted by Hospitalized with Symptoms networks. In fact, there is a correspondence among three communities: (i) Basilicata and Piemonte, (ii) Marche, Emilia, Lombardi, (iii) Veneto and Liguria, Lazio, Toscana. This means that those regions that form the three communities present the same behavior according to the number of the hospitalized patients with symptoms and the total number of hospitalized patients.

Figure 6 shows the evolution of the Home Isolation network over five weeks. In the first week, Lombardi, Veneto and Emilia each shows a different trend of the number of subjects in isolation at home, both between them and compared to other regions. Next, in the network formed by considering the all weeks, Lombardi, Veneto, Emilia and Marche formed a subgraph disconnected by a different dense subgraph composed by the rest of the regions. However, Veneto represents a single community in Figure 13d. This means that the behavior of Veneto presents a low similarity with respect to Lombardi, Emilia and Marche despite forming a module.

Figure 7 and Figure 8 show the Total Currently Positive network and New Currently Positive network. Both networks evolve over five weeks by forming a structure in which all nodes are connected. According to the extracted communities, four modules are identified in Total Currently Positive network (see Figure 13e) and they are formed by: (i) Piemonte; (ii) Lombardi, Veneto, Marche, Emilia, (iii) Basilicata, Molise, Calabria, Sardegna, (iv) Puglia, Friuli, Valle d’Aosta, Toscana, Lazio, Abruzzo, Umbria, Campania, Trento, Liguria Bolzano, Sicilia. The communities identified in New Currently Positive network in Figure 13f are: (i) Piemonte, Marche, Toscana; (ii) Lombardi, Veneto, Emilia, (iii) Basilicata, Molise, (iv) Puglia, Friuli, Valle d’Aosta, Lazio, Abruzzo, Umbria, Campania, Trento, Liguria Bolzano, Calabria, Sardegna, Sicilia.

It is possible to notice that Lombardi, Veneto, Emilia form a community in both Total Currently Positive network and New Currently Positive network, Piemonte represents a single community in the Total Currently Positive network, while Piemonte is associated with Marche and Toscana in the New Currently Positive network.

Figure 9 represents the Discharged/Healed network over five weeks. In the first week, the network structure presents all nodes connected except for Veneto. In the second week, the Discharged/Healed network is formed by three subgraphs; in the third and fourth weeks the network is very dense; in the fifth week, the network structure is characterized by different disconnected components, and finally, at the end of 35 days the network is composed by a subgraph composed by Lombardi and Veneto and another subgraph highly connected. This means that Lombardi and Veneto have a similar behavior that is different from the rest of the Italian regions. Also, Lombardi and Veneto represent one of the five communities extracted by Discharged/Healed network. The extracted communities are reported in Figure 13g.

Figure 10 shows the evolution of Deceased network. The evolution of this network is different from other Italian COVID-19 networks. In fact, in the first week all nodes are disconnected, so all Italian regions present different trends. In the second week, it is possible to notice that Emilia and Marche nodes are disconnected and there is a subgraph composed by Lombardi and Veneto and then there is a large subgraph formed by other regions. In the third week, all nodes present connections. In the fourth and fifth week the Deceased network presents different disconnected components; then, the final network shows a single disconnected node that represents Basilicata. Also, the Basilicata represents a single community of Deceased network; see Figure 13h. The other extracted communities are: (i) Piemonte, Toscana Liguria, Lazio, Friuli, Puglia, Valle d’Aosta, (ii) Lombardi, Veneto, Emilia, Marche, (iii) Sicilia, Molise, Abruzzo, Umbria, Campania, Trento, Bolzano, Calabria, Sardegna.

Figure 11 represents the Total Cases network over five weeks. The final network demonstrates that the Italian regions present a significant level of similarity respect to the number of total coronavirus cases because all nodes are connected. Figure 13i shows the communities identified in Total Cases. The first community is composed by Lombardi, Veneto, Emilia, Marche; the second community is composed by Piemonte; the third community is composed by Basilicata, Molise; the fourth community is composed by Toscana Liguria, Lazio, Campania, Friuli, Sicilia Puglia, Valle d’Aosta formed, whereas Abruzzo, Umbria, Trento, Bolzano, Calabria, Sardegna formed the fifth community.

Figure 12 shows the evolution of the Swab Network that represents the number of performed swab tests. The network, in the first week, shows Lombardi and Veneto nodes disconnected by other regions. In fact, these ones are the Italian regions that initially performed high number of test swabs. Also, the Veneto region has no connections in the final network and this reflects the policy of Veneto to carry out swab tests on asymptomatic subjects, i.e., it is an outlier with respect to other regions. Figure 13j shows the extracted communities in the Swab network. The first community is composed by Veneto; the second community is composed by Lombardi, Emilia; the third community is composed by Basilicata, Molise; the fourth community is formed by Marche, Toscana, Lazio, Piemonte, Friuli, Valle d’Aosta; the fifth community is formed by Sicilia, Campania, Liguria, Puglia; while Abruzzo, Umbria, Trento, Bolzano, Calabria, Sardegna formed the sixth community.

We want to evaluate: (1) if different data present similar or dissimilar communities and (2) if the communities are similar or dissimilar considering different temporal interval on the same data. The Figure 14, Figure 15, Figure 16, Figure 17, Figure 18, Figure 19, Figure 20, Figure 21, Figure 22 and Figure 23 report the evolution of the communities related to different data.

Figure 14 reports the evolution of Hospitalized with Symptoms Network Communities.

Figure 14a reports six communities extracted in the first week: (i) Lombardi, (ii) Veneto, (iii) Emilia, Marche, (iv) Liguria, Toscana, Piemonte, (v) Puglia, Lazio, Campania, Abruzzo, Bolzano, Sicilia, (vi) Umbria, Sardegna, Calabria, Molise, Valle d’Aosta, Friuli, Basilicata, Trento.

At the end of three weeks, Veneto, after representing a community in the previous week, moves in another community, whereas Emilia leaves the community with Marche and becomes a single community. Also, some regions migrate from fifth and sixth communities to other communities. Therefore, Figure 14b reports the five extracted communities after three weeks: (i) Lombardi, (ii) Emilia, (iii)Veneto, Marche, Piemonte, Liguria, Toscana, Lazio, (iv) Trento, Bolzano, Abruzzo, Friuli, Sicilia, Puglia, (v) Campania, Umbria, Sardegna, Calabria, Molise, Valle d’Aosta, Basilicata. Finally, Figure 14c reports five communities in the study period: the first one consists of Basilicata, which leaves the previous community and becomes a single one; the second one is composed by Piemonte, Marche, Emilia, Lombardi, Veneto; the third one is composed by Liguria, Lazio, Toscana; the fourth one is formed by Campania, Puglia, Sicilia, Abruzzo, Valle d’Aosta, Friuli, Trento, Bolzano; the last one is composed by Umbria, Sardegna, Calabria, Molise.

Figure 15 reports the evolution of Intensive Care Network Communities. It is possible to notice that in the first week (Figure 15a), there is: one large community formed by Umbria, Lazio, Piemonte, Toscana Campania, Sicilia, Sardegna, Abruzzo, Umbria, Calabria, Basilicata, Bolzano, Valle d’Aosta, Friuli, Trento, Molise; two single communities formed by Lombardi and Emilia; and a small community formed by Emilia and Marche.

After three weeks the number of extracted communities increases; see Figure 15b. In fact, Lombardi, Sardegna, Valle d’Aosta represent three single communities, Veneto and Emilia form a community, as well as, Marche and Piemonte. Then, Liguria, Lazio and Toscana form a six community, and the last two are composed by (i) Umbria, Campania, Molise, Abruzzo, Friuli, Trento, Puglia and (ii) Calabria, Sicilia, Bolzano, Basilicata.

Finally, in the study period, five communities are mined (see Figure 15c), formed by (i) Lombardi and Veneto, (ii) Liguria and Lazio, (iii) Marche, Emilia, Piemonte, Toscana and (iv) a large module formed by Campania, Sicilia, Sardegna, Abruzzo, Umbria, Calabria, Basilicata, Bolzano, Valle d’Aosta, Friuli, Trento, Molise.

Figure 16 reports the evolution of Total Hospitalized Network Communities.

Figure 16a shows the six mined communities. The first community is composed by Liguria, Toscana, Piemonte, the second one is formed Sardegna, Umbria, Calabria, Basilicata, Valle d’Aosta, Friuli, Trento, Molise; the third module comprises Lazio, Campania, Sicilia, Abruzzo, Bolzano, Puglia; the fourth community is represented by Marche, Emilia; the fifth is represented by Lombardi and the last one consists of Veneto.

After three weeks (see Figure 16b) the regions move, with the exception of Lombardi, which continues to represent a single community and the communities become eight. In fact, Emilia becomes a single community; Veneto becomes a community among with Piemonte and Marche; Toscana moves in the community with Liguria; Lazio and Campania forms a new community, as well as, Basilicata and Valle d’Aosta; another community is formed by: Abruzzo, Puglia, Sicilia and the last one is formed by Friuli, Bolzano, Trento, Umbria, Sardegna, Calabria, Molise.

At the end of the study period, the five communities reported in Figure 16b are formed. The first one consists of Basilicata that leaves the previous community and becomes a single one; the second one is composed by Toscana, Liguria and Lazio; Sardegna, Calabria, Molise leaves the previous large community and form a smaller one; by the fourth one is formed by Piemonte, Marche, Emilia, Lombardi, Veneto; the last one is composed by Umbria, Puglia, Sicilia, Abruzzo, Valle d’Aosta, Friuli, Trento, Bolzano.

Figure 17 reports the evolution of Home Isolation Network Communities.

Figure 17a reports the mined communities at the end of first week. It is possible to notice that Lombardi, Veneto and Emilia form single communities, and then there are three large communities: the first one is represented by Piemonte, Liguria, Marche, Sicilia, Campania; the second one is composed by Puglia, Valle d’Aosta, Toscana, Umbria, Calabria; the third one Trento, Lazio, Abruzzo, Sardegna, Bolzano, Basilicata and Molise, Friuli, Campania.

At the end of three week, Lombardi, Veneto and Emilia move together to form a unique community, whereas, the other regions form new communities, such as (i) Puglia, Trento, Lazio, Umbria, (ii) Calabria, Abruzzo, Valle d’Aosta, Sardegna, Bolzano, Basilicata and Molise, (iii) Sicilia, Toscana, Friuli, Piemonte, Liguria, Marche, Campania (see Figure 17b).

Figure 17c shows the community topology in the study period. Veneto leaves the community among with Lombardi and Emilia and it becomes a single one, whereas, Lombardi, Emilia forms a new module among with Marche. Basilicata and Molise move together to form a unique community. Sardegna, Calabria, Abruzzo, Bolzano form a fourth community. The fifth community is composed by Puglia, Trento, Lazio, Umbria, Sicilia, and the sixth one is represented by Toscana, Piemonte, Valle d’Aosta, Friuli, Liguria, Campania.

Figure 18 reports the evolution of Total Currently Positive Communities. At the end of first week, there are eight mined communities, and they are reported in Figure 18a. The first community is composed by Umbria, Sardegna, Basilicata, Molise, Friuli Toscana, Calabria, Valle d’Aosta, Trento; the second one is formed by Bolzano, Lazio, Abruzzo, Puglia; the third module comprises Campania, Sicilia Liguria; Piemonte and Lazio represent the fourth community; the fifth one consists of Marche; the sixth community is represented by Emilia; the seventh is represented by Lombardi and the last one consists of Veneto.

After three weeks, the number of communities (see Figure 18b) decreases. In fact, it is possible to notice five subgraphs. Emilia joints with Veneto and Lombardi remains single community. Lazio, Sicilia, Friuli, Puglia form a new community; Piemonte, Toscana, Campania, Marche, Liguria, represent a fourth community; Trento, Abruzzo, Umbria, Calabria, Sardegna, Basilicata, Molise, Bolzano, Campania, Valle d’Aosta, form a fifth community;

In the study period, the number of extracted communities further decreases, see Figure 18c The first community is composed by Veneto, Lombardi, Emilia, Marche; the second community is composed by Piemonte; the third community is composed by Basilicata, Molise, Calabria, Sardegna; the fourth community is formed by Toscana, Lazio, Friuli, Valle d’Aosta, Sicilia, Campania, Liguria Puglia, Abruzzo, Umbria, Trento, Bolzano.

Figure 19 reports the evolution of New Currently Positive Communities. Figure 19a reports the mined communities at the end of first week. It is possible to notice that Lombardi, Veneto and Emilia form single communities, and then there are two large communities: the first one is represented by Marche, Piemonte, Liguria, Campania, Abruzzo, Toscana; the second one is composed by Puglia, Valle d’Aosta, Umbria, Calabria, Sicilia, Campania, Trento, Lazio, Sardegna, Bolzano, Basilicata and Molise, Friuli.

After three weeks, there remain five extracted communities but the regions that form them vary; see Figure 19b. The first community is composed by Lombardi; the second community is composed by Veneto and Emilia; the third community is composed by Basilicata, Molise, Valle d’Aosta, Sardegna, Campania, Bolzano; the fourth community is formed by Marche, Toscana, Piemonte; the fifth community is formed by Sicilia, Liguria, Puglia, Abruzzo, Umbria, Trento, Calabria, Lazio, Friuli.

In the study period, the number of extracted communities further decreases, see Figure 19c and there are: (i) Piemonte, Marche, Toscana; (ii) Lombardi, Veneto, Emilia, (iii) Basilicata, Molise, (iv) Puglia, Friuli, Valle d’Aosta, Lazio, Abruzzo, Umbria, Campania, Trento, Liguria Bolzano, Calabria, Sardegna, Sicilia.

Figure 20 reports the evolution of Discharged/Healed Network Communities. It is possible to notice that in the first week (Figure 20a), there are: a large community formed by Umbria, Piemonte, Toscana Campania, Sicilia, Sardegna, Abruzzo, Umbria, Calabria, Basilicata, Bolzano, Emilia, Valle d’Aosta, Friuli, Trento, Molise; a small community formed by Lombardi, Marche, Lazio; and single communities formed by Veneto.

After three weeks the number of extracted communities increases; see Figure 20b. In fact, Lombardi leaves the previous community and becomes a single one; Lazio, Emilia, Liguria and Veneto get together to form a second community; the third is composed by Friuli, Sicilia, Toscana; the last one is formed by Sardegna, Valle d’Aosta, Marche and Piemonte, Umbria, Campania, Molise, Abruzzo, Trento, Puglia, Calabria, Bolzano, Basilicata.

Finally, Figure 20c shows the communities in the study period. It is possible to notice five communities. The first one consists of Lombardi and Veneto; the second one is composed by Emilia, Liguria, Lazio; the third one is composed by Friuli, Campania, Toscana, Sicilia; the fourth one is formed by, Puglia, Abruzzo, Trento; the last one is composed by Umbria, Sardegna, Calabria, Piemonte, Marche, Valle d’Aosta, Bolzano, Basilicata, Molise.

Figure 21 reports the evolution of Deceased Communities. Figure 21a reports the mined communities at the end of first week. It is possible to notice that there is: one large community formed by Emilia, Piemonte, Liguria, Campania, Abruzzo, Puglia, Valle d’Aosta, Umbria, Calabria, Sicilia, Campania, Trento, Lazio, Sardegna, Bolzano, Basilicata and Molise, Friuli; two single communities represented by Lombardi and Veneto; and a single community composed by Marche and Toscana.

After three weeks, the number of extracted communities increases. In fact, the regions that form them vary by forming new communities; see Figure 21b. The first community is composed by Lombardi that remains a single community; the second community is composed by Veneto and the third one is composed by Emilia; the fourth community is formed by Basilicata, Molise, Sardegna, Calabria; the fifth community is formed by Sicilia, Liguria, Puglia, Abruzzo, Umbria, Trento, Lazio, Friuli, Valle d’Aosta, Campania, Bolzano; Marche, Toscana, Piemonte.

In the study period, the number of extracted communities decreases (see Figure 21c) and there are: (i) Basilicata represents a single community, (ii) Piemonte, Toscana Liguria, Lazio, Friuli, Puglia, Valle d’Aosta, (iii) Lombardi, Veneto, Emilia, Marche, (iv) Sicilia, Molise, Abruzzo, Umbria, Campania, Trento, Bolzano, Calabria, Sardegna.

Figure 22 reports the evolution of Total Cases Network Communities.

Figure 22a shows the five mined communities. The first community is composed by Liguria, Toscana Lazio, Piemonte, Campania, Sicilia; the second one is formed Sardegna, Abruzzo, Umbria, Calabria, Basilicata, Bolzano, Valle d’Aosta, Friuli, Trento, Molise, Puglia; the third module comprises Marche, Emilia; the forth one Lombardi and the last one Veneto.

After three weeks (see Figure 22b) there are six communities. Veneto becomes a community among with Emilia; Marche moves in the community with Liguria, Toscana Lazio, Piemonte, Campania; and three new modules are formed: the first one is composed by Sicilia, Friuli and Puglia, the second one is composed by Abruzzo, Bolzano, Trento and Umbria, the third one is formed by Basilicata, Sardegna, Valle d’Aosta, Calabria, Molise.

At the end of the study period, there are five communities extracted. Figure 22c reports the communities. The first community is composed by Lombardi, Veneto, Emilia, Marche; the second community is composed by Piemonte; the third community is composed by Basilicata, Molise; the fourth community is composed by Toscana Liguria, Lazio, Campania, Friuli, Sicilia Puglia, Valle d’Aosta formed a community, whereas Abruzzo, Umbria, Trento, Bolzano, Calabria, Sardegna formed the fifth community.

Figure 23 reports the evolution of the Swab Network Communities. At the end of first week, there are eight mined communities, and they are reported in Figure 23a. The first community is composed by Umbria, Calabria, Valle d’Aosta, Trento, Sicilia, Abruzzo, Puglia; the second one is formed by Sardegna, Basilicata, Molise, Bolzano; the third module comprises Friuli, Campania, Liguria; Piemonte represents the fourth community; the fifth one consists of Toscana and Lazio; the sixth community is represented by Emilia and Marche; the seventh is represented by Lombardi and the last one consists of Veneto.

After three weeks, the number of communities (see Figure 23b) decreases. In fact, it is possible to notice six subgraphs. Emilia leaves the previous community and forms a single one. Lombardi and Veneto join together. Toscana and Lazio continue to form a community; Piemonte, Friuli, Campania, Marche, Puglia, Liguria, Sicilia form a fourth community; Trento, Abruzzo, Umbria, Calabria, form a fifth community and the last one is composed by Sardegna, Basilicata, Molise, Bolzano, Campania, Valle d’Aosta.

In the study period, there remain six extracted communities but the regions that form them vary; see Figure 23c. The first community is composed by Veneto; the second community is composed by Lombardi, Emilia; the third community is composed by Basilicata, Molise; the fourth community is formed by Marche, Toscana, Lazio, Piemonte, Friuli, Valle d’Aosta; the fifth community is formed by Sicilia, Campania, Liguria Puglia; while Abruzzo, Umbria, Trento, Bolzano, Calabria, Sardegna formed the sixth community.

By analyzing the results, it is possible to demonstrate that the topology of the communities varies, i.e., the regions join and leave them along time and the community consistency changes along time on the same data. For the communities related to the different available data, it is possible to notice that after the first week, the extracted communities are different. This changes, after analyzing the communities after three weeks. In fact, the Total Currently Positive Network Communities and New Currently Positive Network show similar communities as well as Deceased Network and Total Cases Network. Finally, after five weeks, the topology of communities is different for all Italian COVID-19 networks except for the Hospitalized with Symptoms Network and Total Hospitalized Network, which show similar extracted communities.

In the literature, there are different works that apply graph theory to analyze the COVID-19 pandemic spread. For example, Reich et al. [13] modeled the COVID-19 spread by using a SEIR (Susceptible–Exposed–Infectious–Recovered–Susceptible) agent-based model on a graph, which takes into account several important real-life attributes of COVID-19: super-spreaders, realistic epidemiological parameters of the disease, testing, and quarantine policies. The agent is represented as a node in a graph, and infection between contacts is represented by graph edges. Then, the authors have applied the SEIR model to analyze the disease progression. Herrmann et al. [14] modeled the human interaction according to three different networks, i.e., Scale-free, Mitigation Hub, and Mitigation Random, and they applied the SIS (Susceptible–Infected–Susceptible) model. The authors demonstrated that network topology could improve the predictive power of SIR model of COVID-19 by providing novel insights into the potential strategies and policies for mitigating and suppressing the spread of the virus.

Kuzdeuov et al. [15] implemented a network-based stochastic epidemic simulator that models the movement of a disease through the SEIR states of a population. The nodes of the networks represent an administrative unit of the country, such as a city or region, and the edges between nodes represent transit links of roads railways, and air travel routes to model the mobility of inhabitants among cities. In [16], Kumar presents a network-based model for predicting the spread of COVID-19, incorporating human mobility through knowledge of migration and air transport.

The work of Wang et al. [17] applied statistical and network analysis on heterogeneous network containing patients and hospitals as nodes and relationships between relatives, friends or colleagues as edges. Network analysis provided important information about patients, hospitals and their relationships and it was able to provide a guidance for the distribution of epidemic prevention materials.

In summary, different works rely on network-based representation for the application of predictive models, whereas only Wang et al. [17] uses statistical and network-based analysis to evaluate an infected cluster of people in different hospitals. To the best of our knowledge, our work is the first study that provides a network-based representation and visualization of COVID-19 data at the regional level and applies network-based analysis to discover communities of regions that show similar behavior.

In conclusion, with this study, we wanted to give a graph-based representation of the COVID-19 measures considering how the regions behaved differently with respect to ten different datasets provided by Italian Civil Protection. It emerged that the regions where the epidemic had a greater impact, such as the Lombardi, Veneto, Piemonte and Emilia, had a different behavior with respect to other regions. This is evident in the community detection in which the regions most affected by the epidemic form individual communities or they are part of the same community. In addition, this study also led to identifying similar behaviors of regions that are geographically distant but that together form community. An example is represented by Calabria, Sardegna, and Molise that represent a cluster in Hospitalized with Symptoms Network, Total Hospitalized Network, Total Currently Positive Network, Discharged/Healed Network, Total Cases, Deceased Network, Intensive Care Network. This can lead the search for indicators that unite the regions such as factors, age structure, health care facilities, and socioeconomic status. Moreover, our visual representation of data can lead the search for indicators that are responsible for community formation i.e., factors common to regions such as age structure, health care facilities, and socioeconomic status. Furthermore, starting from the regions that form communities, it could be possible to plan common interventions such as the increase in intensive care units or the increase in swab tests.

## 5. Conclusions

The COVID-19 disease has spread worldwide in a matter of weeks. In Italy, the epidemic of COVID-19 started in the north and quickly involved all regions. In this paper, we evaluated the evolution of Italian COVID-19 data provided daily by Italian Civil Protection. The main goal of this work is the network-based representation of COVID-19 diffusion similarity among regions and graph-based visualization with the aim of underlining similar diffusion regions. We identified similar Italian regions with respect to the available COVID-19 data and we mapped these in different networks. Finally, we performed a network-based analysis to discover communities of regions that show similar behavior. For future work, we plan to extend the study by considering the evolution of the communities at greater time intervals to demonstrate a new pattern of regions with respect to COVID-19 data.

## Figures and Tables

**Figure 1 ijerph-17-04182-f001:**
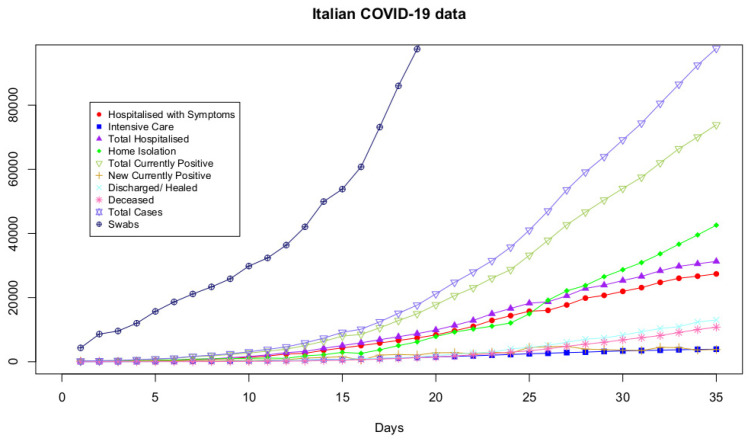
The trends of Hospitalized with Symptoms data, Intensive Care data, Total Hospitalized data, Home Isolation data, Total Currently Positive data, New Currently Positive data, Discharged/Healed data, Deceased data, Total Cases data, Swabs data. Day 1 is 24 February 2020.

**Figure 2 ijerph-17-04182-f002:**
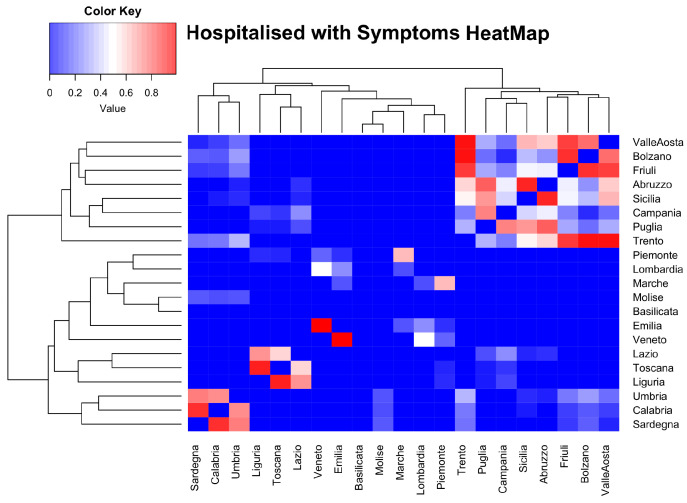
The figure shows the heat map related to results obtained by applying Wilcoxon Sum Rank test in the study period on Hospitalized with Symptoms data.

**Figure 3 ijerph-17-04182-f003:**
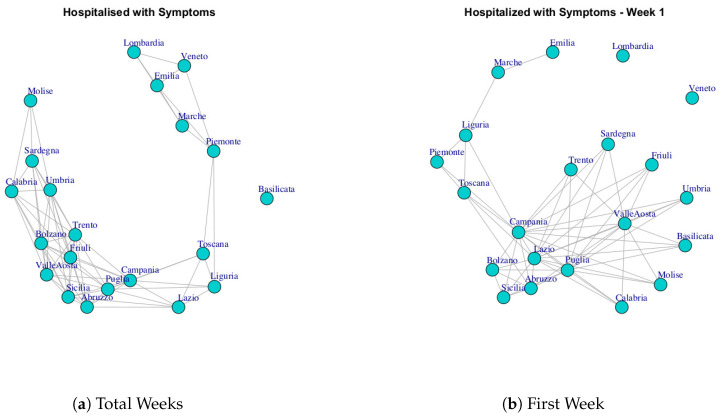
Evolution of Hospitalized with Symptoms Network.

**Figure 4 ijerph-17-04182-f004:**
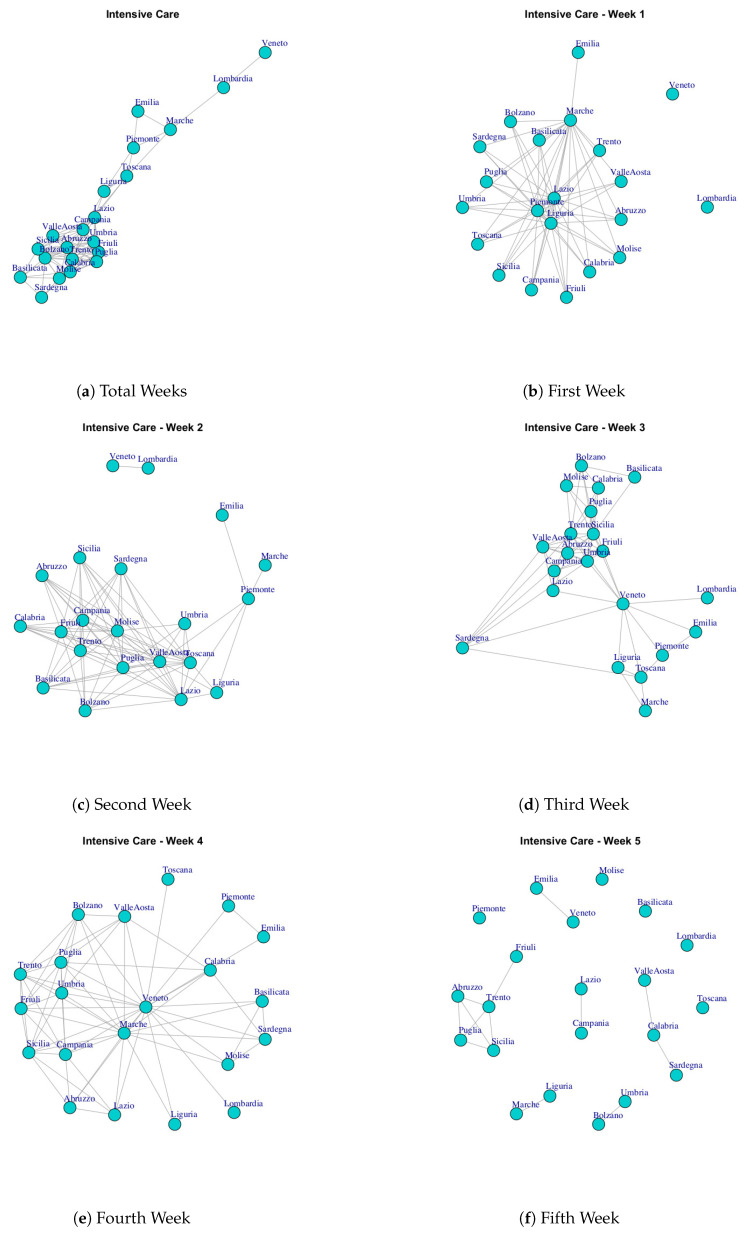
Evolution of Intensive Care Network.

**Figure 5 ijerph-17-04182-f005:**
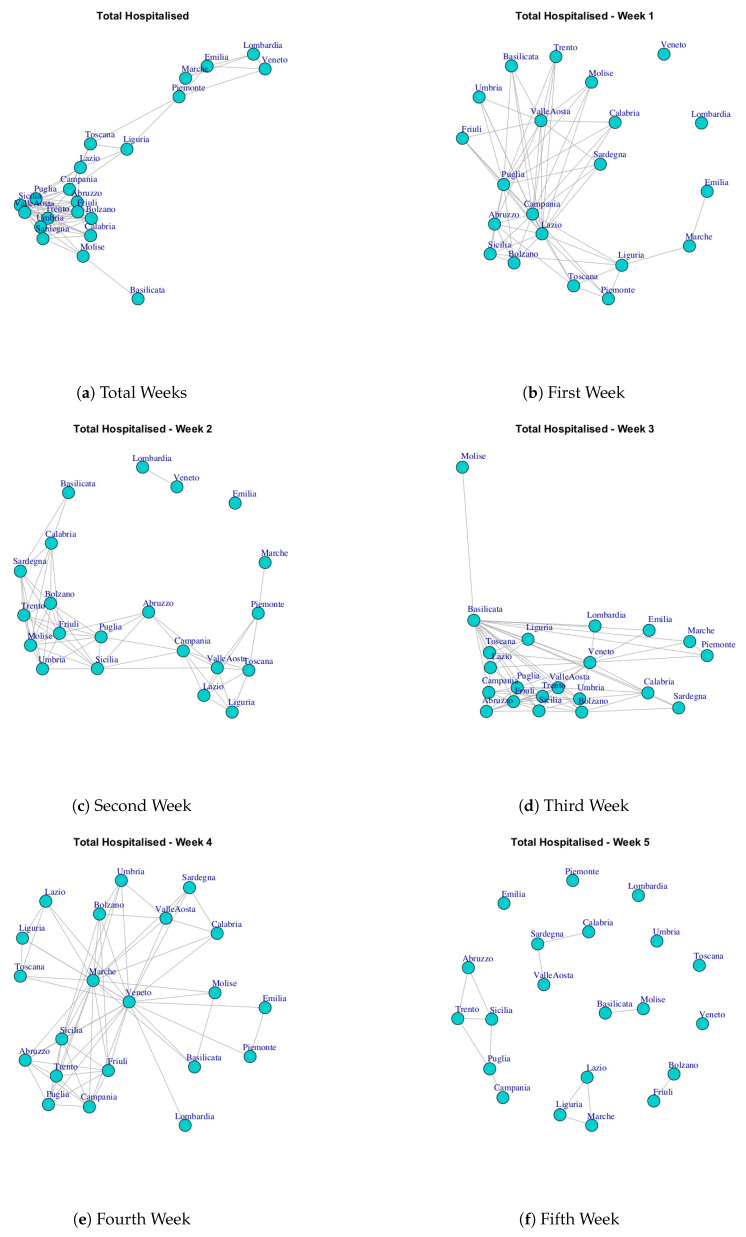
Evolution of Total Hospitalized Network.

**Figure 6 ijerph-17-04182-f006:**
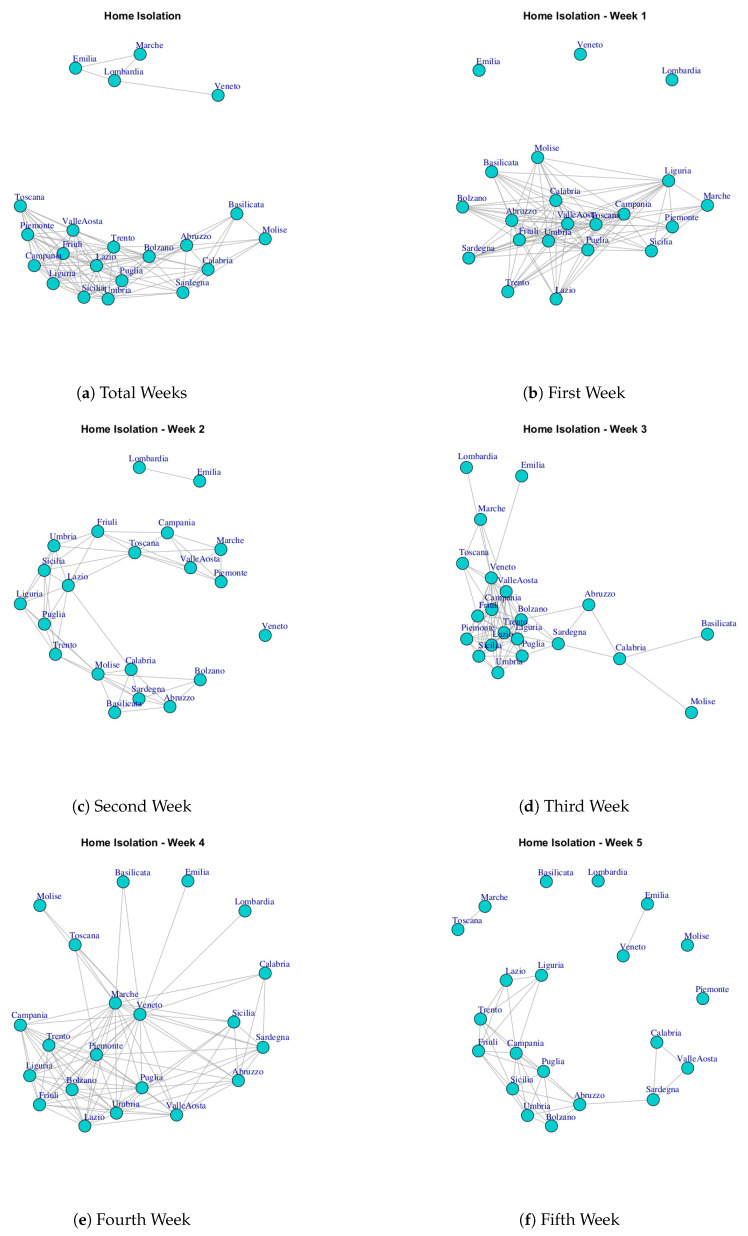
Evolution of Home Isolation Network.

**Figure 7 ijerph-17-04182-f007:**
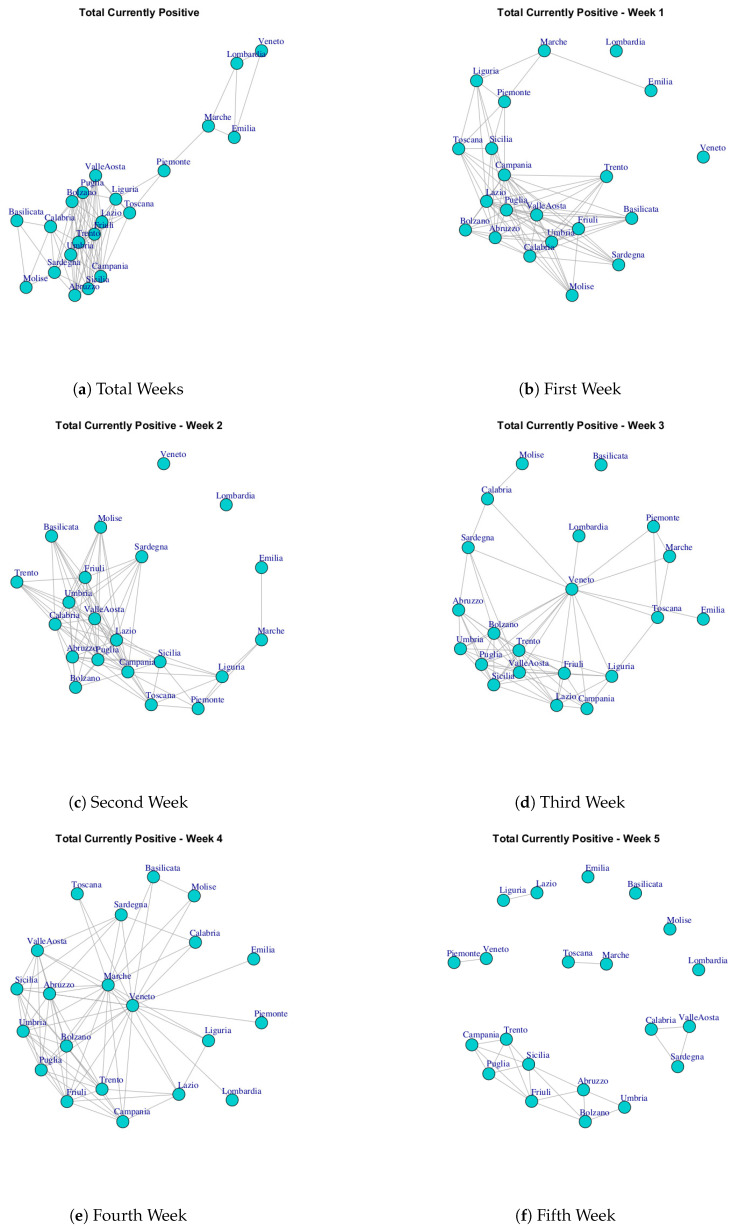
Evolution of Total Currently Positive Network.

**Figure 8 ijerph-17-04182-f008:**
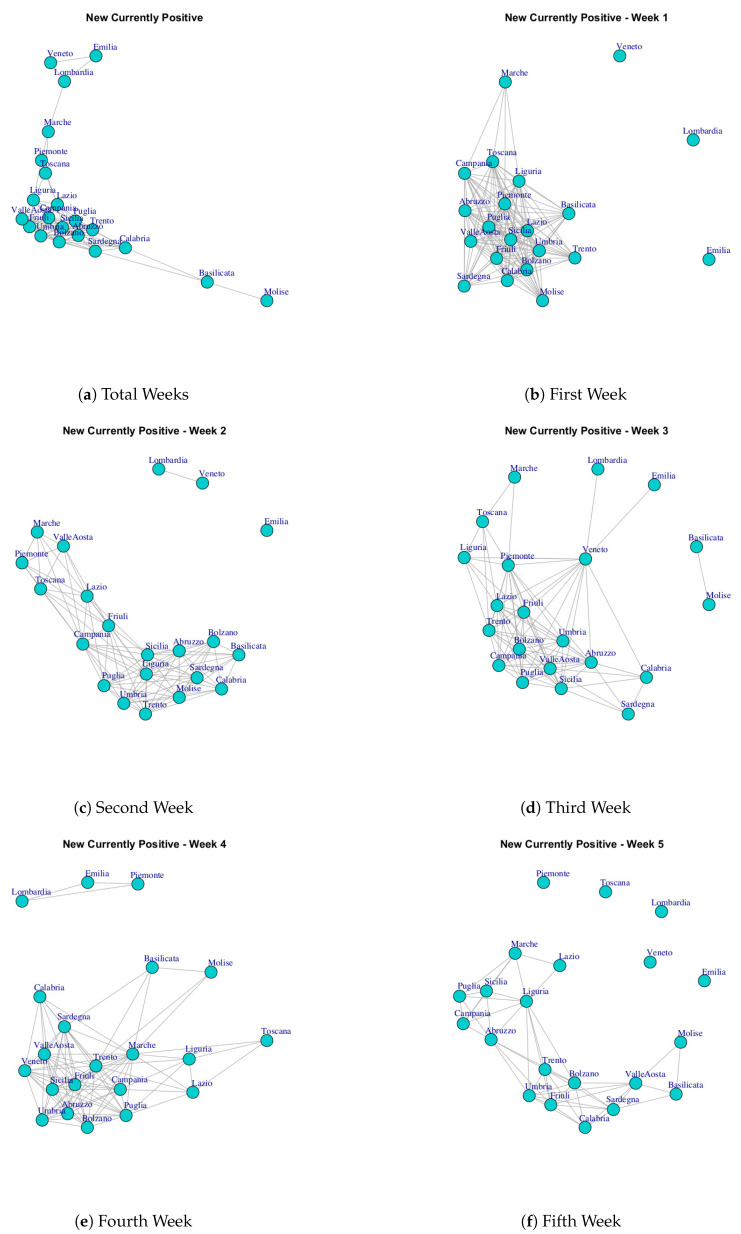
Evolution of New Currently Positive Network.

**Figure 9 ijerph-17-04182-f009:**
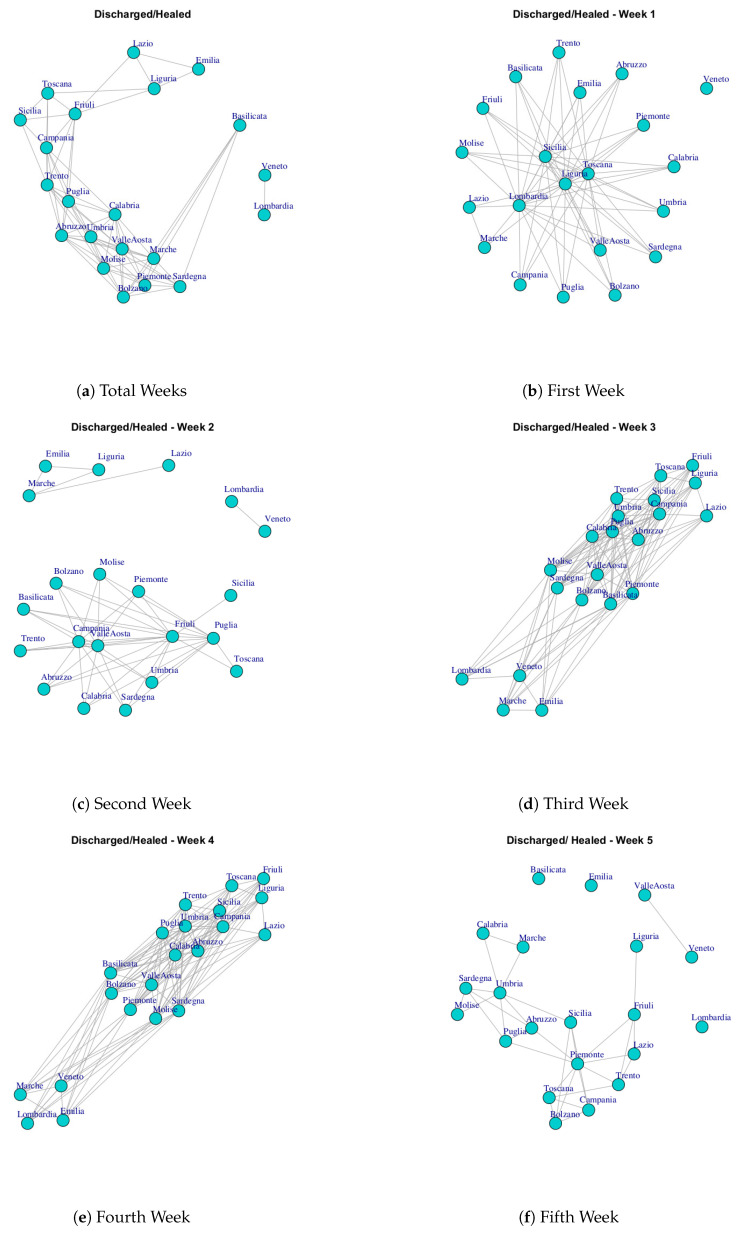
Evolution of Discharged/Healed Network.

**Figure 10 ijerph-17-04182-f010:**
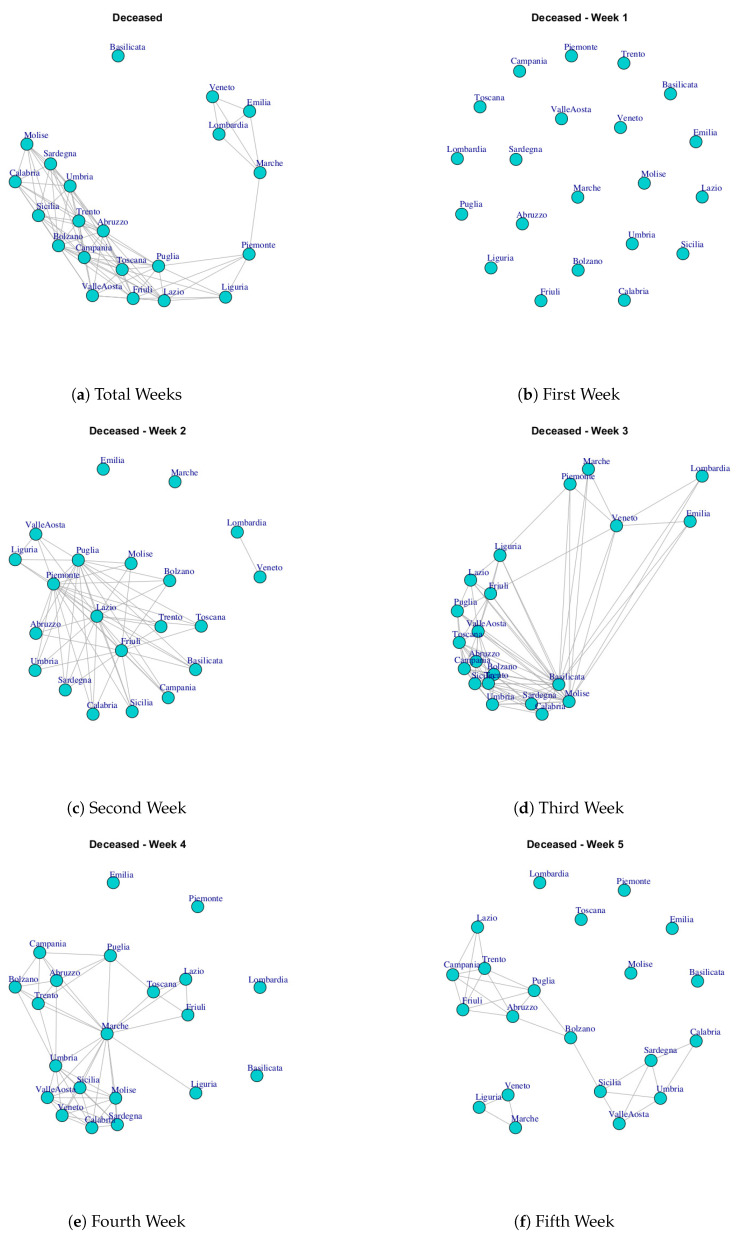
Evolution of Deceased Network.

**Figure 11 ijerph-17-04182-f011:**
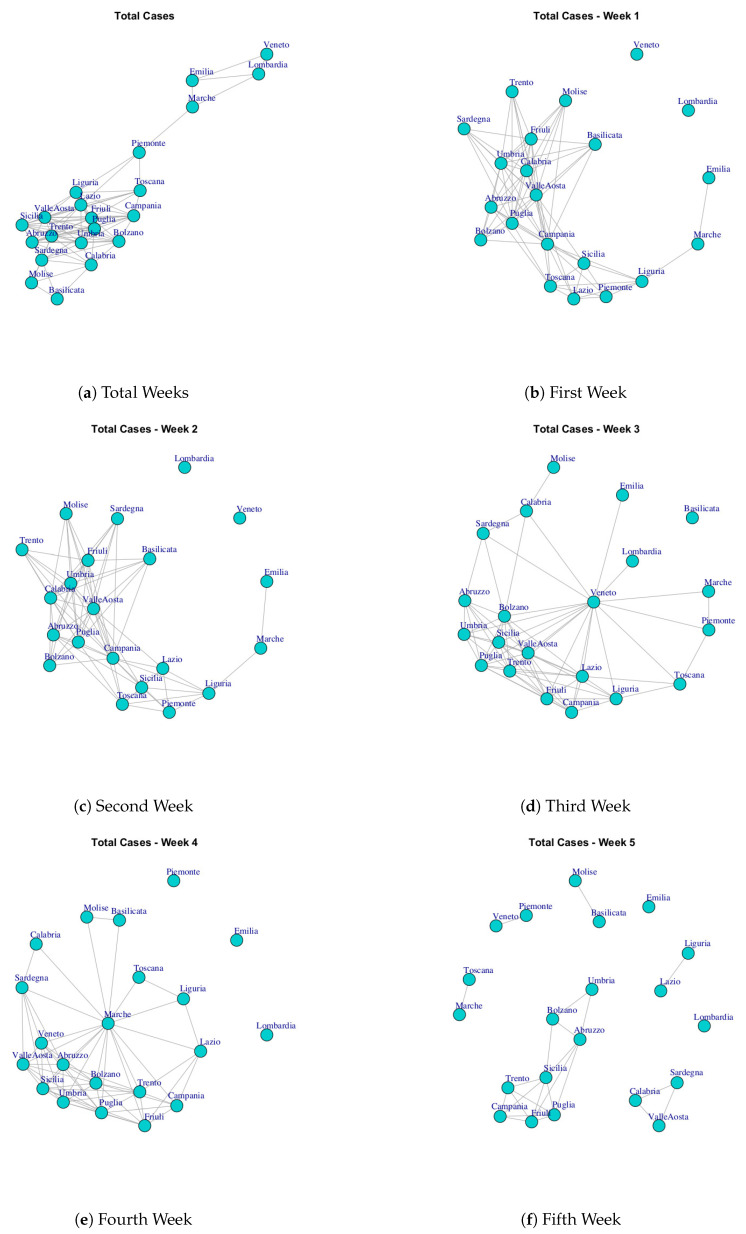
Evolution of Total Cases Network.

**Figure 12 ijerph-17-04182-f012:**
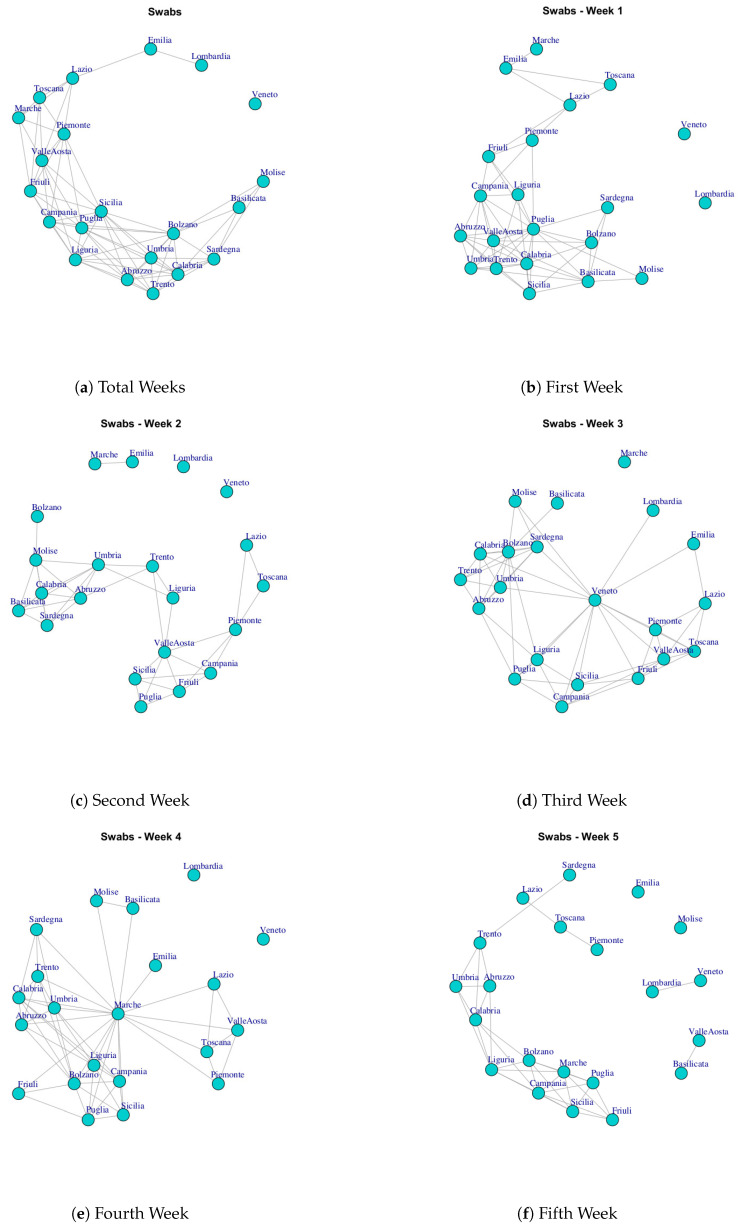
Evolution of Swabs Network.

**Figure 13 ijerph-17-04182-f013:**
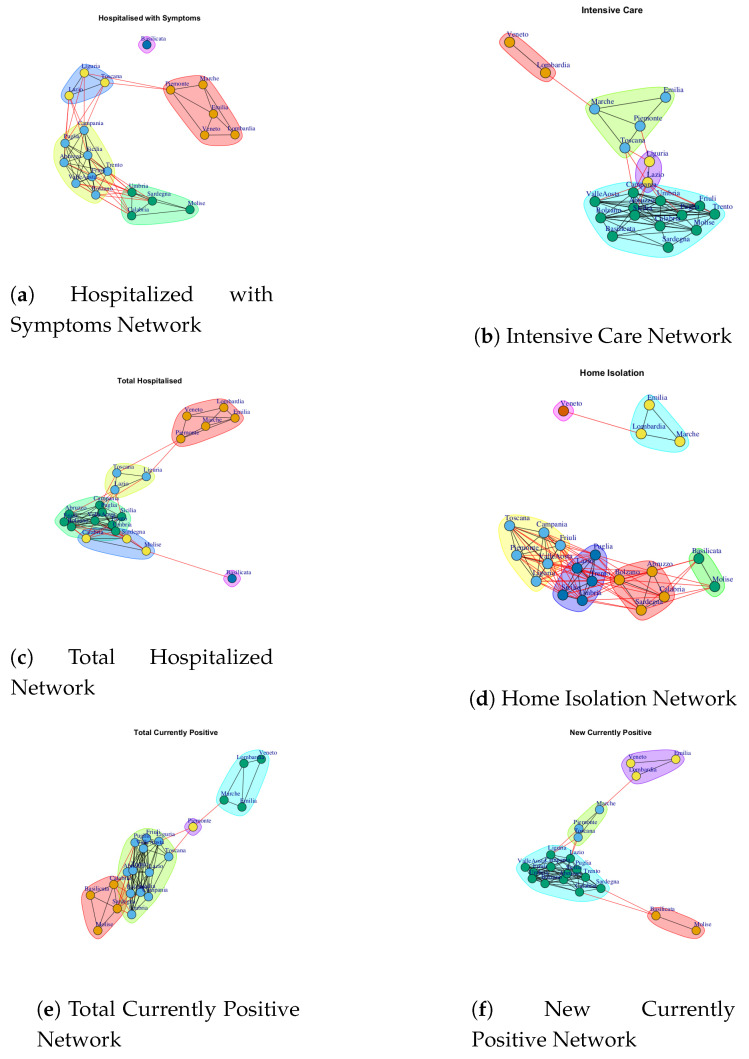
The figure shows the communities identified in Italian COVID-19 networks in the study period.

**Figure 14 ijerph-17-04182-f014:**
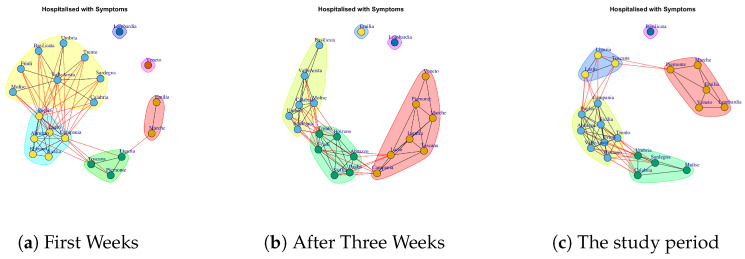
Evolution of Hospitalized with Symptoms Network Communities.

**Figure 15 ijerph-17-04182-f015:**
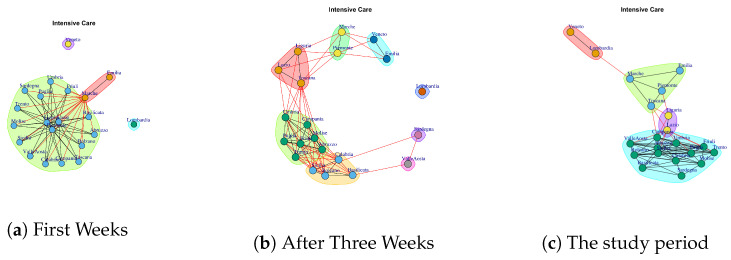
Evolution of Intensive Care Network Communities.

**Figure 16 ijerph-17-04182-f016:**
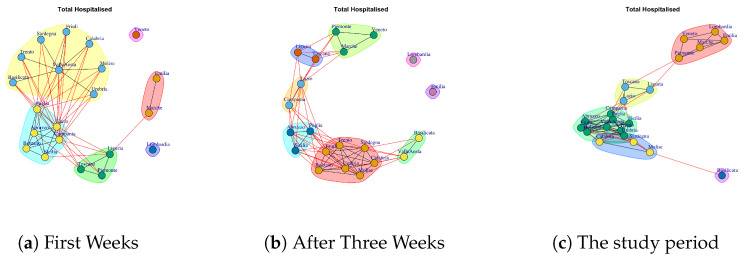
Evolution of Total Hospitalized Network Communities.

**Figure 17 ijerph-17-04182-f017:**
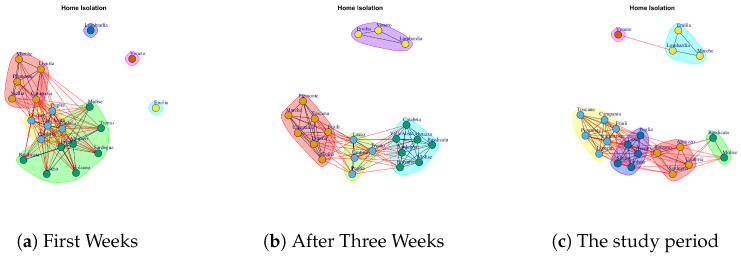
Evolution of Home Isolation Network Communities.

**Figure 18 ijerph-17-04182-f018:**
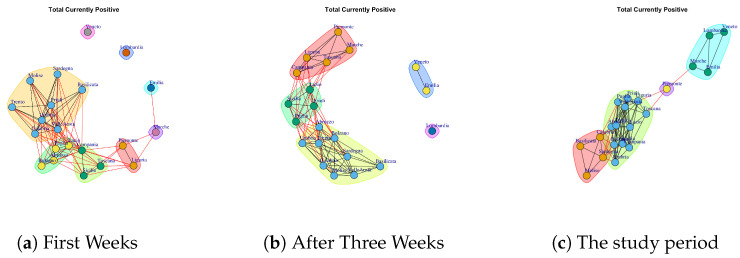
Evolution of Total Currently Positive Network Communities.

**Figure 19 ijerph-17-04182-f019:**
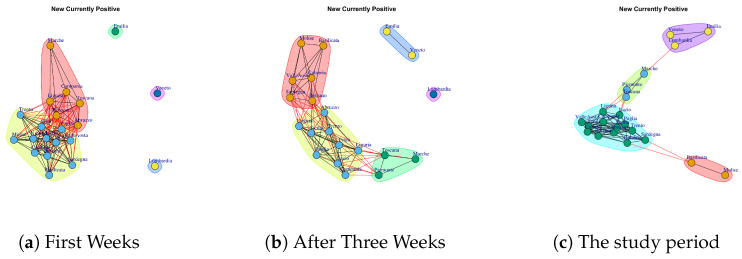
Evolution of New Currently Positive Network Communities.

**Figure 20 ijerph-17-04182-f020:**
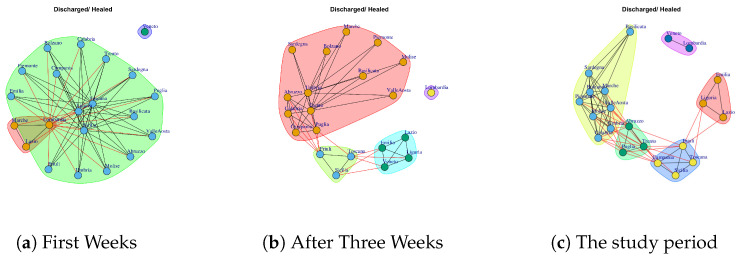
Evolution of Discharged/Healed Network Communities.

**Figure 21 ijerph-17-04182-f021:**
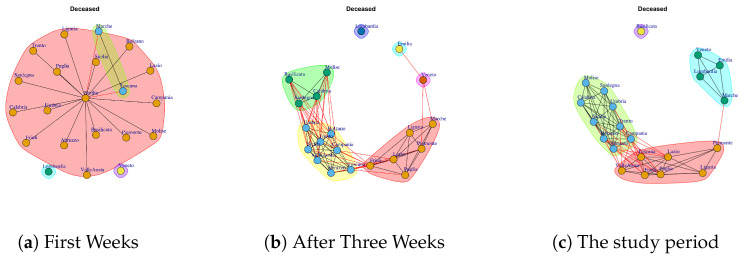
Evolution of Deceased Network Communities.

**Figure 22 ijerph-17-04182-f022:**
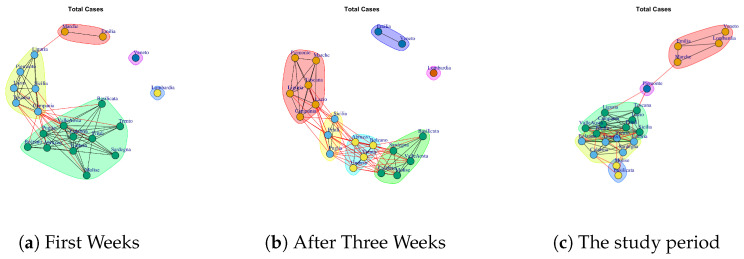
Evolution of Total Cases Network Communities.

**Figure 23 ijerph-17-04182-f023:**
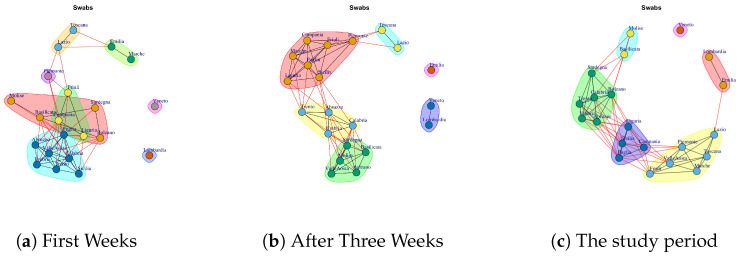
Evolution of Swabs Network Communities.

**Table 1 ijerph-17-04182-t001:** Descriptive statistics for all regions in the study period.

Region	Statistics	Hospitalized with Symptoms	Intensive Care	Total Hospitalized	Home Isolation	Total Currently Positive	New Currently Positive	Discharged/Healed	Deceased	Total Cases	Swabs
Abruzzo	sample size	35	35	35	35	35	35	35	35	35	35
	mean	198.968	36.079	235.048	664.429	899.476	45.381	105.397	102.571	1107.444	10,235.143
	sd	144.026	26.174	167.182	654.035	797.205	38.981	135.614	107.148	1026.840	11,061.168
	median	280	41	344	516	860	40	23	63	946	5488
	Q1	31	9	26	11	37	7	0	1	38	310
	Q3	324	57	375	1309.5	1679.5	68.5	185.5	202	2067	18,764.5
Basilicata	sample size	35	35	35	35	35	35	35	35	35	35
	mean	27.254	7.905	35.159	96.714	131.873	5.810	18.381	8.016	158.270	2407.730
	sd	25.359	6.897	30.277	84.957	114.971	6.862	33.324	9.511	144.778	2857.527
	median	22	7	37	95	133	3	0	1	134	1046
	Q1	1	0.5	2	5.5	7.5	1	0	0	7.5	151.5
	Q3	57.5	14.5	65.5	180.5	246	9	14	16	310	3873
Bolzano	sample size	35	35	35	35	35	35	35	35	35	35
	mean	130.000	26.063	156.063	592.143	748.206	39.381	221.730	95.270	1065.206	11,782.746
	sd	101.958	22.913	124.171	505.379	603.912	35.063	299.997	101.007	940.736	12,108.597
	median	146	20	176	525	791	35	67	48	906	7744
	Q1	12.5	2.5	15	41.5	56.5	5	0	0	56.5	55.5
	Q3	224	45	271	955.5	1275	57	400.5	195.5	1956	21,526
Calabria	sample size	35	35	35	35	35	35	35	35	35	35
	mean	87.111	9.349	96.460	309.889	406.349	17.286	38.492	30.222	475.063	9352.778
	sd	67.007	7.393	72.266	280.259	347.032	19.409	56.738	31.717	422.581	9747.839
	median	101	8	124	248	372	13	7	14	393	5933
	Q1	9	2	11	4.5	14	2	1	0	16	382.5
	Q3	153	15	160.5	606	788.5	26.5	53.5	65.5	908	17,065
Campania	sample size	35	35	35	35	35	35	35	35	35	35
	mean	322.016	59.381	381.397	1077.333	1458.730	68.746	211.857	122.238	1792.825	18,071.143
	sd	249.885	48.701	288.747	995.719	1262.173	58.133	306.788	125.413	1633.241	20,305.951
	median	448	58	562	631	1169	61	58	83	1310	8346
	Q1	48	9.5	58.5	83.5	137.5	18	2.5	0.5	140.5	1258
	Q3	555	100	647	2301	2929.5	98	262.5	234.5	3479.5	32,763
Emilia	sample size	35	35	35	35	35	35	35	35	35	35
	mean	2249.873	223.810	2473.683	5045.413	7519.095	388.095	2054.937	1348.746	10,922.778	52,816.127
	sd	162.468	28.585	188.984	673.216	843.362	40.047	137.073	109.628	1077.360	11,368.142
	median	2846	269	3122	5195	8850	350	792	1174	10,816	42,395
	Q1	707	101	808	694.5	1502.5	206	34.5	99	1636	6067
	Q3	3490.5	329.5	3823.5	9392	13,049.5	547.5	3520	2439	19,381.5	88,821.5
Friuli	sample size	35	35	35	35	35	35	35	35	35	35
	mean	116.841	25.476	142.317	619.778	762.095	46.302	424.492	98.746	1285.333	17,036.603
	sd	78.685	20.716	98.349	480.525	561.032	38.233	501.817	93.203	1087.776	17,808.502
	median	140	24	163	688	911	41	197	72	1223	10,721
	Q1	20.5	5.5	27	83	110	14	6.5	4.5	121	1837.5
	Q3	177.5	42.5	218	1123	1321	66.5	799	182	2371	28,891
Lazio	sample size	35	35	35	35	35	35	35	35	35	35
	mean	737.000	104.825	841.825	1152.556	1994.381	100.127	371.492	142.333	2508.206	36,767.063
	sd	566.172	82.268	647.447	1108.178	1733.590	68.227	432.751	139.154	2283.257	37,361.801
	median	878	113	991	844	1835	99	155	106	2096	63
	Q1	61	16.5	75	37	112	28	15	6	133	3591
	Q3	1254	186.5	1454.5	2229.5	3681.5	157	703.5	268	4653	63,505
Liguria	sample size	35	35	35	35	35	35	35	35	35	35
	mean	616.730	92.984	709.714	1069.889	1779.603	118.857	716.048	384.111	2879.762	11,983.048
	sd	448.690	62.407	508.265	1000.597	1428.992	80.617	881.009	378.883	2602.383	12,820.717
	median	761	102	874	875	2027	122	260	280	2567	7304
	Q1	67	31.5	97	57.5	154.5	42.5	5	8	167.5	859.5
	Q3	1027	149	1178	2106.5	3317	184	1254	721.5	5283.5	20,201
Lombardi	sample size	35	35	35	35	35	35	35	35	35	35
	mean	7544.905	842.306	8372.113	10,128.694	18,500.806	1160.694	8877.210	5497.323	32,875.339	11,1489.758
	sd	4403.531	443.647	4865.273	8412.344	12,634.922	671.258	7941.673	4831.205	25,165.184	97,350.121
	median	9266	935.5	10,479	9787	21,390	1157.5	7560	4667.5	33,617.5	84,689.5
	Q1	3585.5	513	4098.5	1299.5	5126.5	780	898	542.5	6535.5	23,554
	Q3	11,740.5	1219	13,004	16,754.5	29,894	1549.5	16,551.5	10,374.5	56,820	191,313.5
Marche	sample size	35	35	35	35	35	35	35	35	35	35
	mean	612.333	103.746	840.095	1718.381	2558.476	110.762	859.492	557.111	3975.079	18,888.698
	sd	378.014	94.687	1151.257	3262.427	4400.857	122.858	3100.359	1666.182	9097.186	43,559.955
	median	742	106	872	1652	2795	92	9	310	3114	8623
	Q1	182	56	242	179	421	47.5	0	15.5	436.5	1546.5
	Q3	947.5	140.5	1078	2300.5	3230.5	139	1188.5	685.5	5147.5	19,515
Molise	sample size	35	35	35	35	35	35	35	35	35	35
	mean	17.365	4.778	32.429	113.873	146.302	5.476	49.302	21.508	217.111	2093.238
	sd	12.129	7.349	89.911	324.486	413.371	8.942	240.871	109.997	762.507	6553.059
	median	21	4	29	46	81	3	14	8	103	670
	Q1	4	2	6.5	8	15	0	0	0	15.5	229
	Q3	27	6	33.5	161	193	6	38	13.5	244.5	2135
Piemonte	sample size	35	35	35	35	35	35	35	35	35	35
	mean	2004.476	246.460	2205.762	4051.143	6256.905	387.778	1145.937	830.667	8233.508	33,106.302
	sd	1412.884	167.620	1589.582	4300.008	5660.761	277.052	1749.509	914.100	8134.870	38,023.991
	median	2633	293	2925	2631	5556	490	75	449	6024	16,655
	Q1	312.5	70.5	383	75.5	458	71.5	0	19	477	2402.5
	Q3	3285	389	3613	8018	11,873	590	2054.5	1582.5	15,510	60,017
Puglia	sample size	35	35	35	35	35	35	35	35	35	35
	mean	339.349	50.921	428.397	1033.302	1461.698	68.349	223.746	163.587	1849.032	17,381.683
	sd	264.156	47.506	454.876	1681.225	2109.020	62.838	819.326	364.631	3249.349	22,393.542
	median	464	55	517	610	1095	70	22	65	1182	9191
	Q1	33	3.5	38	25	63	12	1	4	68	828
	Q3	593	74.5	668	1663.5	2369	97	242	245.5	2856.5	28,637.5
Sardegna	sample size	35	35	35	35	35	35	35	35	35	35
	mean	66.063	14.333	86.254	392.190	478.444	20.746	80.698	38.302	597.444	6286.794
	sd	48.419	11.939	80.488	404.945	483.277	21.714	127.515	58.416	647.390	8671.555
	median	90	19	110	350	462	14	13	19	494	3461
	Q1	9.5	0	9.5	19	28.5	3	0	0	28.5	243.5
	Q3	110.5	24	134.5	693.5	820	30.5	124	71	1077	9782
Sicilia	sample size	35	35	35	35	35	35	35	35	35	35
	mean	291.683	36.825	328.508	717.778	1046.286	48.492	117.032	75.238	1238.556	18,631.270
	sd	235.386	28.512	259.761	664.821	906.990	39.304	156.846	82.626	1119.763	20,974.102
	median	346	39	414	658	1095	45	36	33	1164	9658
	Q1	21	1.5	21.5	49	70.5	16.5	2	0	72.5	1074.5
	Q3	523.5	62.5	570.5	1359	1984	70.5	198	151	2333	32,471.5
Toscana	sample size	35	35	35	35	35	35	35	35	35	35
	mean	614.079	157.556	771.635	2347.889	3119.524	145.190	384.857	244.825	3749.206	38,949.286
	sd	443.947	106.565	548.316	2159.489	2600.205	102.694	596.281	261.543	3318.192	41,645.213
	median	791	182	959	1677	2973	151	95	158	3226	20,952
	Q1	95.5	47	136	151	287	54.5	4	1	292	2688.5
	Q3	1006.5	253.5	1258.5	4656.5	5907	221.5	475	460.5	6842.5	73,878.5
Trento	sample size	35	35	35	35	35	35	35	35	35	35
	mean	194.635	36.540	231.175	794.032	1025.206	61.810	364.571	140.905	1530.683	8887.032
	sd	141.717	29.422	170.261	682.862	828.585	49.140	511.393	147.494	1397.661	9902.902
	median	235	38	279	728	1094	64	117	86	1297	4600
	Q1	23.5	3.5	27	35	62	9.5	2.5	0	64.5	463
	Q3	330.5	65.5	390	1539.5	1872.5	97.5	584.5	279.5	2893	15,813.5
Umbria	sample size	35	35	35	35	35	35	35	35	35	35
	mean	83.111	24.143	107.254	304.810	412.063	21.714	266.556	25.857	704.476	9377.429
	sd	63.988	17.534	81.024	255.039	334.552	26.258	349.658	24.549	574.593	10,190.101
	median	97	24	121	282	407	9	12	20	802	5428
	Q1	7.5	3.5	63	29.5	40.5	2	1	0	41.5	300
	Q3	140.5	41	180.5	557	737.5	32	516	52	1305.5	16,993
ValleAosta	sample size	35	35	35	35	35	35	35	35	35	35
	mean	59.381	10.635	70.016	238.905	308.921	17.556	122.937	50.651	482.508	1841.619
	sd	45.414	9.612	53.000	190.234	242.055	20.339	190.354	52.707	422.643	1872.617
	median	71	9	89	280	375	7	2	28	408	1203
	Q1	3	0	0	16	18	1	0	0.5	18.5	94
	Q3	97.5	20	116.5	432.5	548.5	29	187.5	107	890.5	3396
Veneto	sample size	35	35	35	35	35	35	35	35	35	35
	mean	933.127	22.127	103.540	295.921	399.460	19.635	266.048	26.000	691.508	9239.238
	sd	625.491	19.072	83.886	261.567	343.965	25.194	350.048	24.410	586.796	10,305.636
	median	1189	22	121	278	407	7	12	20	802	5428
	Q1	233	67.5	300.5	561	861.5	119.5	50.5	27.5	939.5	19,021.5
	Q3	1456	283	1748	8361	9945	415	2084.5	812	13,594.5	185,806

**Table 2 ijerph-17-04182-t002:** Multiple linear regression Results.

Regions	Population Density People per km${}^{2}$	Intensive Care Beds
Abruzzo	121	73
Basilicata	56	49
Bolzano	71	48
Calabria	128	107
Campania	424	350
Emilia	199	650
Friuli	153	494
Lazio	341	540
Liguria	286	75
Lombardi	422	1067
Marche	162	400
Molise	69	30
Piemonte	172	320
Puglia	206	320
Sardegna	68	150
Sicilia	194	441
Toscana	162	447
Trento	79	75
Umbria	104	69
ValleAosta	39	15
Veneto	267	498

**Table 3 ijerph-17-04182-t003:** *p*-values associated with the Population Density and Intensive Care Beds and the Multiple R-squared.

	Hospitalized with Symptoms	Intensive Care	Home Isolation	Total Currently Positive	Discharged/Healed	Deceased	Total Cases	Swabs
Population Density	*p*-value > 0.05	*p*-value > 0.05	*p*-value > 0.05	*p*-value > 0.05	*p*-value > 0.05	*p*-value > 0.05	*p*-value > 0.05	*p*-value > 0.05
Bed	*p*-value > 0.05	*p*-value > 0.05	*p*-value > 0.05	*p*-value > 0.05	*p*-value > 0.05	*p*-value > 0.05	*p*-value > 0.05	*p*-value > 0.05
R^2^	0.617	0.631	0.685	0.666	0.504	0.544	0.627	0.318

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
