# Peer review of "Statistical and Network-Based Analysis of Italian COVID-19 Data: Communities Detection and Temporal Evolution"

_ijerph, 2020, doi:10.3390/ijerph17124182_

Round 1
Reviewer 1 Report
The paper deals with Coronavirus disease outbreak and presents an analysis of different data at the regional level for the period February 24 to March 29, 2020. The methodology analysis included: Building several similarity matrices based statistical test comparing on similarity or dissimilarity of regions with respect to the ten types of available data. Network-based analysis was used for discovering communities of regions that show similar behavior. Network analysis was performed by running several community detection algorithms on those networks and by underlying communities of regions with similar picture. The network-based analysis of Italian COVID-19 data sophisticatedly showed how regions form communities along time and how a community changes in time and with respect to the different available data.
This a nice piece of work with contemporary epidemic/public health, but also methodological interest. It is, in general, well written and comprehensible in all parts, and thus it is worth publishing.
While the paper explain in detail the findings, in my opinion, what is missing is a general discussion about the theoretical and practical implications of the findings. I suggest that the authors add a paragraph in the final discussion section, elaborating more this issues, that is, what helpful information we get about the how the virus is spreading (e.g., is it an underlying model?), and what advice could be given for prevention actions.
Author Response
Reviewer #1 (Remarks to the Author):
The paper deals with Coronavirus disease outbreak and presents an analysis of different data at the regional level for the period February 24 to March 29, 2020. The methodology analysis included: Building several similarity matrices based statistical test comparing on similarity or dissimilarity of regions with respect to the ten types of available data. Network-based analysis was used for discovering communities of regions that show similar behavior. Network analysis was performed by running several community detection algorithms on those networks and by underlying communities of regions with similar picture. The network-based analysis of Italian COVID-19 data sophisticatedly showed how regions form communities along time and how a community changes in time and with respect to the different available data. This a nice piece of work with contemporary epidemic/public health, but also methodological interest. It is, in general, well written and comprehensible in all parts, and thus it is worth publishing.
While the paper explain in detail the findings, in my opinion, what is missing is a general discussion about the theoretical and practical implications of the findings.
I suggest that the authors add a paragraph in the final discussion section, elaborating more this issues, that is, what helpful information we get about the how the virus is spreading (e.g., is it an underlying model?), and what advice could be given for prevention actions.
Answer: We thanks the reviewer to pointing out this. We added this aspect at LINES 441-458 adding the following paragraphs:
“In conclusion, with this study we wanted to give a graph-based representation of the COVID-19 data considering how the regions behaved differently with respect to ten different data provided by civil protection.
It emerged that the regions where the epidemic had a greater impact such as the Lombardia, Veneto, Piemonte and Emilia had a different behaviour with respect to other regions. This is evident in the community detection in which these regions formed a single community or a community among them. In addition, this study also led to identifying similar behaviors of regions that are geographically distant but that together form community. An example is represented by Calabria, Sardinia and Molise, three far regions, that represent a cluster in Hospitalized with Symptoms Network, Total Hospitalized Network, Total Currently Positive Network, Discharged / Healed Network, Total Cases, Deceased Network, Intensive Care Network.
This can lead for the search for indicators that unite the regions such as factors, age structure, health care facilities, and socioeconomic status.
Moreover, our visual representation of data can lead to the search for indicators that are responsible for community formation i.e. factors common to regions such as age structure, health care facilities, and socioeconomic status.
Furthermore, starting from the regions that form communities, it could be possible to plan common intervention such as the increase in intensive care units or the increase in swab tests.”
Reviewer 2 Report
The authors studied the evolution of COVID-19 from February 24 to March 29, 2020, in 21 regions of Italy. Although the study is of scientific interest, the reviewer has some concerns as follows:
- All the eight indicators you explored in your analyses and the derived estimates are based on the crude data. It is plausible that some major characteristics of the 21 regions may vary in terms of geographic factors, population density, age structure, health care facilities, and socioeconomic status, etc. These factors may have some impacts on the outcomes you studied. I was wondering if it is possible that you perform standardization of the eight indicators using major confounding factors, after then, you explore the evolution with the standardized data but not the crude data.
- Table 1 should present the major characteristics of the 21 regions and the summary of the eight indicators but not the correlation matrices. The current Table 1 has some redundant data. You may present the correlation coefficients in a heat map.
Author Response
Reviewer #2 (Remarks to the Author):
The authors studied the evolution of COVID-19 from February 24 to March 29, 2020, in 21 regions of Italy. Although the study is of scientific interest, the reviewer has some concerns as follows:
All the eight indicators you explored in your analyses and the derived estimates are based on the crude data. It is plausible that some major characteristics of the 21 regions may vary in terms of geographic factors, population density, age structure, health care facilities, and socioeconomic status, etc. These factors may have some impacts on the outcomes you studied.
I was wondering if it is possible that you perform standardization of the eight indicators using major confounding factors, after then, you explore the evolution with the standardized data but not the crude data.
Answer: We thanks the reviewer to pointing out this. We add multiple linear regression in Statistical analysis section.
We performed multiple linear regression by considering nine indicators: Hospitalised with Symptoms, Intensive Care, Home Isolation, Total Currently Positive, Discharged/ Healed, Deceased, Total Cases, Swabs and two geographic factors: population density and intensive care beds for regions.
We perform a standardization of variables as preprocessing step (LINES 134-150).
Table 1 should present the major characteristics of the 21 regions and the summary of the eight indicators but not the correlation matrices.
The current Table 1 has some redundant data. You may present the correlation coefficients in a heat map.
Answer: We thanks the reviewer to pointing out this. We replace the table with the heat map related Hospedalised with Symptoms data in the study period (LINE 119-123). We present the heat maps related to all Italian Covid-19 data in the Appendix (LINE 507).
Reviewer 3 Report
- The paper deals with important issue and proposed approach is interesting but the manuscript has methodological and technical problems.
- The objectives of the paper must be formulate more clear and specific.
- The same for used algorithms and methods. Paper must be self-explained.
- (lines 51- 56) Why Wilcoxon statistical test was used for comparison of two region? Why was two different region considered as paired? In my opinion they are independent. What‘s sample sizes used for comparison? Why does, for example, p=0.47 indicates more similarity then p=0.38? There is no statistical evidence for this.
- (lines 96 -98) The sentence “As a preliminary test, we applied Pearson’s chi-square test. Since p-value was less than 0.05 for each distribution, we decided to use non-parametric test for the following comparison…” is problematic. By definition of Pearson’s chi-square test checks if two categorical variables are independent and it is non-parametric.
- (line 99- 100) Once more, why “The Wilcoxon test performs a pair-wise comparison among regions with the goal to evidence which ones show different trend”. Why is it used for independent groups (regions) and how can it say something about trends? Need explanation.
- Must be explained more detailed how time trends are analyzed and how does “we analize the evolution of the extracted communities over a period of time”.
- Statistical results must be represented not only by p-value but by relevant statistics (means, medians, percentiles and so on) as well.
- Tables and Figures must be explained more detailed.
- The considered phenomena is multifactorial. Thus it requests multivariate analysis. In my opinion, at least population density of the region must be taken in consideration.
Author Response
Reviewer #3 (Remarks to the Author):
The paper deals with important issue and proposed approach is interesting but the manuscript has methodological and technical problems.
The objectives of the paper must be formulate more clear and specific.
Answer: We apologize since we were not able to clarify the aim of our paper. The main aim of this paper is the evaluation of similarity among Italian Regions with respect to data provided by the Italian Civil Protection through a graphic (network-based) representation and the identification of clusters of regions with similar behaviour.
We re-wrote the objectives of the paper at LINES 32-45 ->
“The aim of this paper consists of providing a graph-based representation of data daily provided by the Italian Civil Protection that enables to evaluate which regions show similar behaviour and discover communities. The data refer to the period February 24 to March 29, 2020.
To do this, we design an analysis pipeline to model Italian COVID-19 data as networks and to perform network-based analysis.
At first, for each type of data, we evaluate the similarity among pair of regions using statistical test and according to this, we built ten similarity matrices (one for each Italian COVID-19 data). After that, we map the similarity matrices into networks where the nodes represent the Italian region, and the edges connect statistically similar regions.
Finally, we evaluate how the networks evolve over the weeks by analyzing the networks at different time points: (i) on the period February 24 to March 29, 2020 (study period) and (ii) on single weeks.
Then, network-based analysis was performed mainly
discovering communities of regions that show similar behaviour.
The main contribution of the paper is a network-based representation of COVID-19 diffusion similarity among regions and graph-based visualization to underline similar diffusion regions.
The same for used algorithms and methods.
Paper must be self-explained.
Answer: We apologize for this. We modified the paper structure and we added a section that presents the pipeline that we designed to analyze the Italian COVID-19 data and we present the application of the analysis pipeline in the result section (LINES 48-169)
(lines 51- 56) Why Wilcoxon statistical test was used for comparison of two region? Why was two different region considered as paired? In my opinion they are independent. What‘s sample sizes used for comparison? Why does, for example, p=0.47 indicates more similarity then p=0.38? There is no statistical evidence for this.
Answer: We thanks the reviewer to pointing out this. We design a methodology in order to perform a network-based analysis on Italian COVID-19 data. For this aim, we want to identify a similarity benchmark for the analyzed data. Once similarity measure is applied to data, we built a similarity matrix , where the (i, j) value of the matrix M for data k represents a value obtained by performing a similarity measure. After that we map the similarity matrix to a network.
So, we used as similarity benchmark the Wilcoxon Rank Sum Test or Mann Whitney Wilcoxon Test , nonparametric tests to compare outcomes between two independent groups. An underlying assumption for appropriate use of the tests described was that the continuous outcome was approximately normally distributed or that the samples were sufficiently large (usually n1> 30 and n2> 30) to justify their use based on the Central Limit Theorem. When comparing two independent samples when the outcome is not normally distributed and the samples are small, a nonparametric test is appropriate. Wilcoxon Rank Sum Test, is used to test whether two samples are likely to derive from the same population (i.e., that the two populations have the same shape). Some investigators interpret this test as comparing the medians between the two populations.
In our analysis, we considered each region as independent sample and we defined the null hypotheses H0 i.e. two samples are equal.
The simple size is equal 35 and we add this information in Table 1 at LINE 109.
We applied Wilcoxon test to perform a pair-wise comparison among regions and when the p-value is greater than 0.05, so we can accept the hypothesis H0 of significant equality among two regions, otherwise there is not significant equality.
We considered the different p value level >0.05 as weight factor for the edges when we built the network from similarity matrix.
(lines 96 -98) The sentence “As a preliminary test, we applied Pearson’s chi-square test. Since p-value was less than 0.05 for each distribution, we decided to use non-parametric test for the following comparison…” is problematic. By definition of Pearson’s chi-square test checks if two categorical variables are independent and it is non-parametric.
Answer: We apologize for this. We re-write the sentence ->
As a preliminary test, we applied Pearson's chi-square test.
The p-value was less than 0.05 for each distribution data, i.e. data was not normally distributed.
According to this, we performed paired comparison and multiple comparison of data by using two non-parametric tests: Wilcoxon Sum Rank test and Kruskal-Wallis test.’ (LINES 109-112).
(line 99- 100) Once more, why “The Wilcoxon test performs a pair-wise comparison among regions with the goal to evidence which ones show different trend”. Why is it used for independent groups (regions) and how can it say something about trends? Need explanation.
Answer: We apologize for this. We re-write the sentence -> We applied the Wilcoxon test to perform a pair-wise comparison among regions with the goal to highlight statistically similar distributions among them. (LINES 117-118)
Must be explained more detailed how time trends are analyzed and how does “we analize the evolution of the extracted communities over a period of time”.
Answer: We apologize since we were not able to clarify this point. At first, we extracted the communities from Italian COVID-19 networks related to the period February 24 to March 29, 2020. (study period) and related to single weeks. After that, we extracted the communities by considering aggregated data, i.e. data at week k, includes all data from week 1 to k. In this way, we want to analyze the evolution of the extracted communities in three time intervals: i) after the first week, after three weeks and after five weeks (the study period).
The aim is to evaluate: 1) if different data present similar or dissimilar communities and 2) if the communities are similar or dissimilar considering different temporal interval on the same data.
We explain these aspects at LINEs (261-268).
Statistical results must be represented not only by p-value but by relevant statistics (means, medians, percentiles and so on) as well.
Answer: We thanks the reviewer to pointing out this. We add a table that summarize the main descriptive statistics (LINE 109).
Tables and Figures must be explained more detailed.
Answer: We thanks the reviewer to pointing out this. We explain in detail the Figures in Discussion Section (LINES 171- 459) and the Tables in Appendix (LINES 479-505)
The considered phenomena is multifactorial. Thus it requests multivariate analysis. In my opinion, at least population density of the region must be taken in consideration.
Answer: We thanks the reviewer to pointing out this. We add multiple linear regression in Statistical analysis section.
We performed multiple linear regression by considering nine indicators: Hospitalised with Symptoms, Intensive Care, Home Isolation, Total Currently Positive, Discharged/ Healed, Deceased, Total Cases, Swabs and two geographic factors: population density and intensive care beds for regions.
We perform a standardization of variables as preprocessing step (LINES 134-150).
Round 2
Reviewer 2 Report
The authors addressed my concerns, I have no further comments.
Author Response
First of all, we thank the reviewers for the time they spent on the revision of our paper.
We addressed all the suggestions of the Editor, and we thank for their comments which we found helpful for refining the manuscript and correcting some remaining errors.
We carefully read the text, fixing several typos and mistakes and we asked a native English speaker to proofread the paper.
The new parts are evidenced in bold in the revised manuscript.